# Evidence of thermophilization in Afromontane forests

Aida Cuni-Sanchez [1,2] ✉, Emanuel H. Martin [3], Eustrate Uzabaho [4], Alain S. K. Ngute [5], Robert Bitariho [6], Charles Kayijamahe[4], Andrew R. Marshall[5,7], Nassoro A. Mohamed [3], Gideon A. Mseja[3], Aventino Nkwasibwe[6], Francesco Rovero [8,9], Douglas Sheil [10], Rogers Tinkasimire[6], Lawrence Tumugabirwe[6], Kenneth J. Feeley [11,12] & Martin J. P. Sullivan [13] ✉

Thermophilization is the directional change in species community composition towards greater relative abundances of species associated with warmer environments. This process is well-documented in temperate and Neotropical plant communities, but it is uncertain whether this phenomenon occurs elsewhere in the tropics. Here we extend the search for thermophilization to equatorial Africa, where lower tree diversity compared to other tropical forest regions and different biogeographic history could affect community responses to climate change. Using re-census data from 17 forest plots in three mountain regions of Africa, we find a consistent pattern of thermophilization in tree communities. Mean rates of thermophilization were +0.0086 °C·y⁻¹ in the Kigezi Highlands (Uganda), +0.0032 °C·y⁻¹ in the Virunga Mountains (Rwanda-Uganda-Democratic Republic of the Congo) and +0.0023 °C·y⁻¹ in the Udzungwa Mountains (Tanzania). Distinct from other forests, both recruitment and mortality were important drivers of thermophilzation in the African plots. The forests studied currently act as a carbon sink, but the consequences of further thermophilization are unclear.

As global temperatures rise, species are predicted and observed to shift their geographical distributions towards cooler latitudes and elevations[1], affecting the taxonomic and functional composition of plant communities[2–4]. Poleward or upward range shifts of species should result in greater relative abundances of more-thermophilic species from relatively warmer climates – a phenomenon referred to as 'thermophilization'[5]. While such a phenomenon is well-documented in temperate and boreal regions, much less information is available about the responses of tropical plant communities, despite tropical plants being especially susceptible to climate change because of their narrow thermal niches[6].

Evidence for thermophilization in tropical forests currently comes from the Neotropics, where changes in the community temperature index (CTI, the community weighted mean of species' thermal optima)

[1]Department of International Environmental and Development Studies (NORAGRIC), Norwegian University of Life Sciences, Ås, Norway. [2]Department of Environment and Geography, University of York, York, UK. [3]College of African Wildlife Management, Mweka, Tanzania. [4]International Gorilla Conservation Programme, Musanze, Rwanda. [5]Forest Research Institute, University of the Sunshine Coast, Sippy Downs, QLD, Australia. [6]Institute of Tropical Forest Conservation, Mbarara University of Science and Technology, Mbarara, Uganda. [7]Flamingo Land Ltd, Malton, UK. [8]Department of Biology, University of Florence, Sesto Fiorentino, Italy. [9]MUSE-Museo delle Scienze, Trento, Italy. [10]Forest Ecology and Forest Management Group, Wageningen University & Research, Wageningen, The Netherlands. [11]Department of Biology, University of Miami, Coral Gables, FL, USA. [12]Fairchild Tropical Botanic Garden, Coral Gables, FL, USA. [13]Department of Natural Sciences, Manchester Metropolitan University, Manchester, UK. ✉e-mail: a.cunisanchez@york.ac.uk; martin.sullivan@mmu.ac.uk

in adult trees were +0.0065 °C yr⁻¹ in Colombia, Peru, Ecuador, Argentina and Costa Rica[7–10] and +0.008 °C yr⁻¹ in Jamaica[11]. Where responses of juvenile trees (<10 cm diameter) have been recorded (Colombia), thermophilzation occurs at similar rates in adult and juvenile trees[8]. In these abovementioned studies, the observed shifts in tree species composition were attributed primarily to increased mortality of species with ranges centred at higher, cooler elevations. In other words, in these neotropical forests species range shifts and associated compositional shifts were occurring mostly via range retractions rather than range shifts or expansions[12]. If generalisable, these findings imply that many tropical tree species have at best a poor capacity to persist under rapidly rising temperatures, suggesting a high risk for local species loss in these biodiverse tropical forest ecosystems[13].

Important questions remain about the generalisability of these results, both within the Neotropics but especially between continents. Tropical forests in Africa are known to be less diverse than in the Neotropics because of lower speciation rates and higher extinction rates driven by aridification over the Cenozoic[14,15]. During the cold arid phases of the Ice Ages, tropical forests in Africa retreated into small refugia, leaving a legacy in the current distribution of individual species as well as patterns of diversity and community composition[15,16]. While some of these forest refugia were located in mountain regions, other mountains lost their forests entirely and were only later recolonised[17]. Also, because of topographic and orographic effects, some of the driest tropical montane forests are found in Africa[18]. We expect this combination of low species diversity and exposure to past climate fluctuations to result in Afromontane forest tree species having relatively broad climate niches (compared to Neotropical species exposed to higher species richness and greater climate stability[19]). Because of these broad ecological niches, tree species in Afromontane forests may be less sensitive to climatic changes than those found in neotropical montane forests. Indeed, a recent study on lowland tropical forests in Africa suggests that tree species in African lowland forests may be more resilient to climatic extremes such as El Niño than those in Amazonian and Asian lowland forests[20]. If Afromontane forests are in fact less sensitive to increased temperatures than Andean montane forests, then species range shifts may be slower, resulting in less thermophilization. Beyond lower tree diversity, Afromontane forests are known to have a different structure to montane forests in the Neotropics, with lower tree density but greater abundance of large stems[21], contributing to the slower dynamics of African forests[22]. This abundance of large stems, with slower demographic processes, might further decrease thermophilzation rates in Afromontane forests or mean it is initially only visible through recruitment and mortality of smaller stems with faster dynamics.

Climate change driven shifts in tree species composition could impact tropical forests' function as aboveground stores and sinks of carbon[23]. If mortality of cooler-affiliated species occurs faster than recruitment of warm-affiliated species, then forest carbon stocks could decline. Alternatively, if recruits from lower elevations are faster-growing and can reach larger sizes then thermophilization could lead to an increase in carbon stocks. Andean forests currently act as a carbon sink[8] despite experiencing thermophilization, but while Afromontane forests are known to have large carbon stocks[21], the dynamics of this carbon are not known.

In this study we investigate the responses of tree communities in tropical montane forests to anthropogenic climate change in three mountain regions of Africa: the Kigezi Highlands (Uganda), the Virunga Mountains (Rwanda-Uganda-DR Congo) and the Udzungwa Mountains (Tanzania) (Fig. 1). Specifically, we test for patterns of compositional change and quantify rates of thermophilization through time. We also assess the relative contributions of tree mortality and recruitment to the observed compositional shifts to gain additional insight into the underlying demographic processes and species' responses and investigate the implications of the findings for forest carbon stocks. We find a consistent pattern of thermophilization in the Afromontane forests studied, driven by both recruitment and mortality. Overall, these forests are carbon sinks even though climate warming appears to be changing tree species' composition. However, the consequences of further thermophilization are unclear, as we discuss below.

## Results

We repeatedly censused 17 1-hectare forest inventory plots located between 610 and 3388 m asl. Each plot was censused three times between 2010 and 2022 (see Methods), by measuring all stems ≥10 cm diameter at breast height. This resulted in a dataset of 26,850 measurements of 9528 stems belonging to 207 species (mean 23.4 species per plot), of which 166 species (93.0% of stems) had sufficient occurrence records to calculate their thermal optima (see Methods). We calculated the community temperature index (CTI, in °C) for each plot during each census as the mean of species' thermal optima (estimated as the average of mean annual temperatures at known occurrence locations) weighted a) by number of individual stems ($CTI_{Stem}$) or b) by the basal-area of each stem ($CTI_{BA}$). $CTI_{Stem}$ is more sensitive to recruitment and turnover of small stems, as all stems are weighted equally, while mortality of large stems would have a greater impact on $CTI_{BA}$.

CTI increased with the mean annual temperature of each plot ($CTI_{Stem}$: 0.45 ± 0.03 SE per °C, $t = 13.0$, df = 15, $P < 0.001$, $R^2 = 0.92$; $CTI_{BA}$: 0.43 ± 0.04 SE per °C, $t = 11.8$, df = 15, $P < 0.001$, $R^2 = 0.90$). These relationships indicate that turnover in species composition along the elevation gradients tracks species' macroscale thermal affiliations, with a slightly stronger association evident for changes in species' occurrences rather than changes in species' basal area. However, for both metrics change in CTI lagged change in temperature, with under half a degree change in CTI for each degree C increase in temperature (Fig. 2).

A stronger signature of temporal change was evident for $CTI_{Stem}$ than $CTI_{BA}$. Thirteen of the 17 plots (76.5%) exhibited increasing $CTI_{Stem}$ over the study period (Fig. 3a), compared to ten plots for $CTI_{BA}$ (Fig. 3b). Three plots (17.6%) decreased in $CTI_{Stem}$ (six plots for $CTI_{BA}$) and one plot that was dominated by a single species exhibited no change for either metric of CTI. Across all plots the mean change in $CTI_{Stem}$ was +0.0045 °C·y⁻¹ (95% confidence intervals = 0.0019–0.0075, $V = 127$, $P = 0.002$, see Supplementary Fig. 1 for consequences of using different thresholds for including species). While $CTI_{BA}$ also increased on average (+0.0020 °C·y⁻¹), this trend was not statistically significant (95% confidence intervals = −0.0013–0.0052, $V = 84$, $P = 0.423$).

Mean rates of change in $CTI_{Stem}$ were fastest at the Kigezi Highlands (0.0086 °C·y⁻¹), followed by Virunga and Udzungwa Mountains (0.0032 and 0.0023 °C·y⁻¹ respectively); however, this variation in thermophilization rates across sites was not statistically significant (Kruskall-Wallis test, $X^2 = 1.16$, $P = 0.559$). Site-level changes in $CTI_{BA}$ were weaker (and negative for Udzungwa) but maintained the same rank order of change as $CTI_{Stem}$. Rates of change in $CTI_{Stem}$ were comparable with rates of mean annual temperature change during the study period (Kigezi = 0.0059 °C·y⁻¹, Virunga = 0.0033 °C·y⁻¹, Udzungwa = 0.019 °C·y⁻¹, based on Chelsa monthly climatologies 2010–2019[24]).

Although changes in CTI were most evident when looking across the entire monitoring period, $CTI_{Stem}$ also increased significantly between the first and second censuses (Wilcoxon signed rank test, $V = 19$, $P = 0.012$) and between the second and third censuses ($V = 13$, $P = 0.005$). Nine plots had increasing $CTI_{Stem}$ across both census intervals, seven had increasing $CTI_{Stem}$ in just one census interval, one plot had no change in either interval; no plots had decreasing $CTI_{Stem}$ in both census intervals (Fig. 3a).

The clearer signal for change in $CTI_{Stem}$ compared to $CTI_{BA}$ supports the role of recruitment and mortality of smaller stems in driving thermophilization, and the inertia in responses of large stems. Some

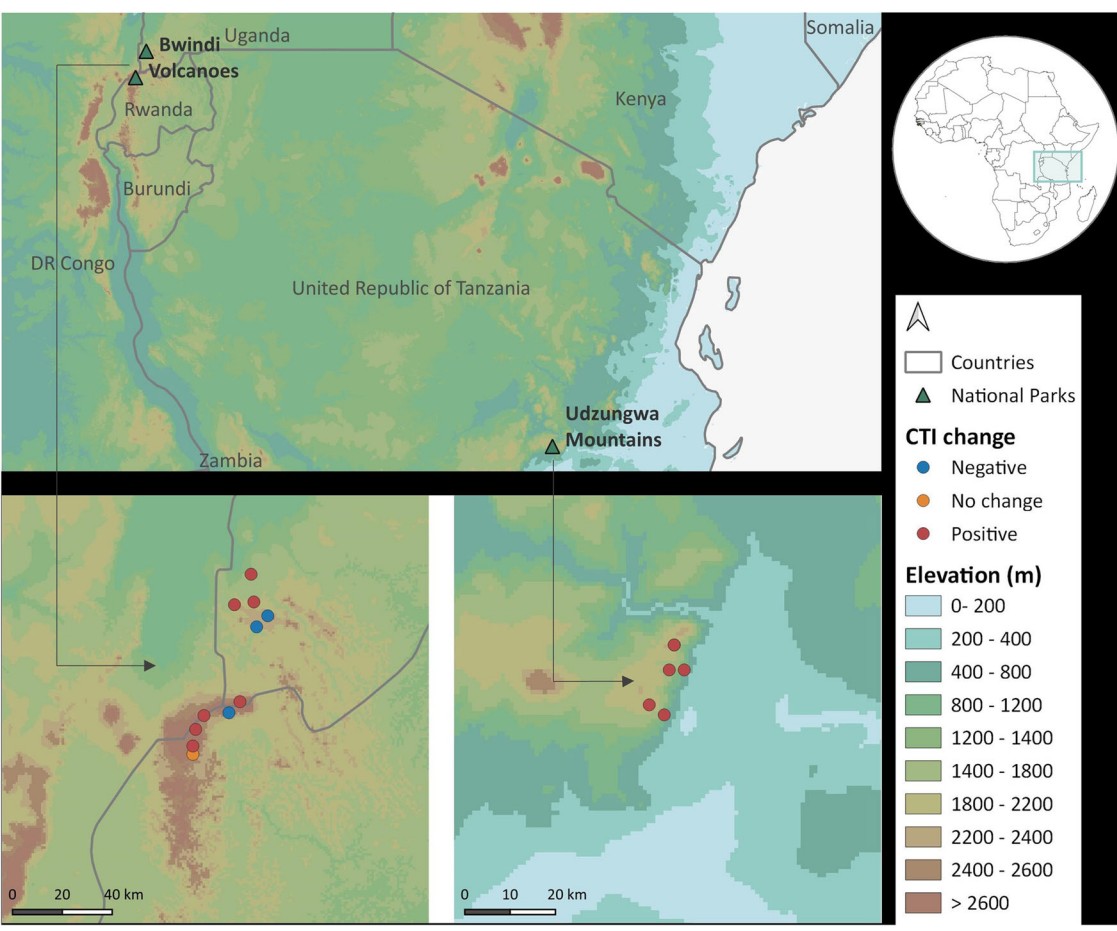

**Fig. 1 | Location of study plots.** Plots were located in Bwindi Impenetrable National Park (the Kigezi Highlands, Uganda $n = 5$ plots), the Volcanoes National Park (Virunga Mountains in Rwanda-Uganda, $n = 6$ plots) and the Udzungwa Mountains National Park (Tanzania, $n = 6$ plots). CTI community temperature index. Figure created using QGIS version 3.28.15. Elevation data from NASA (https://www.un-spider.org/links-and-resources/data-sources/digital-elevation-model-srtm-1-arc-second-30m-nasa-nga.) Country boundaries from ICPAC, accessed through https://open.africa/dataset/africa-shapefiles.

further support for this is provided by partitioning change in $CTI_{Stem}$ into components due to recruitment and mortality. While change in $CTI_{Stem}$ due to mortality was stronger than change in $CTI_{Stem}$ due to recruitment (mortality: $0.003 \pm 0.001\,°C·y^{-1}$, recruitment: $0.001 \pm 0.002\,°C·y^{-1}$), both effects were weaker than the overall trend in $CTI_{Stem}$ indicating that it results from a combination of both processes (Fig. 4a). In contrast, the non-significant trend in $CTI_{BA}$ was driven by mortality, with no signal for thermophilization due to recruitment (Fig. 4b). Thus, while recruitment is contributing to thermophilization, the larger recruits (i.e. faster growing species) do not necessarily have warmer thermal optima.

In contrast to CTI, a community precipitation index (CPI, calculated in the same way as CTI, but using the average of mean annual precipitation at known occurrence locations) was not significantly related to the mean annual precipitation at each plot (slope = $0.251 \pm 0.208$ SE, $t = 1.21$, $P = 0.245$, $R^2 = 0.089$, Supplementary Fig. 2). Neither abundance nor basal-area weighted CPI changed significantly over time (CPI$_{Stem}$: $V = 45$, $P = 0.245$, CPI$_{BA}$: $V = 41$, $P = 0.171$, Supplementary Fig. 2). The weak relationship between CPI and precipitation could reflect the challenges of characterising fine-scale precipitation patterns in African montane regions[25].

Despite the shifts in species composition observed here, the plots studied acted as a carbon sink during the study period (mean net change in aboveground carbon stocks = $0.95$ Mg C ha$^{-1}$ yr$^{-1}$, 95% CI = 0.63–1.27), with a mean increase of $0.17$ Mg C ha$^{-1}$ yr$^{-1}$ due to tree recruitment, $1.93$ Mg C ha$^{-1}$ yr$^{-1}$ due to tree growth, and a mean loss of $1.14$ Mg C ha$^{-1}$ yr$^{-1}$ through tree mortality (Supplementary Fig. 3). The net change in aboveground carbon stocks of each plot was not related to changes in abundance or basal-area weighted CTI (Supplementary Table 1), nor where carbon fluxes due to recruitment, growth and mortality related to the respective partitioned components of CTI change (Supplementary Table 2).

## Discussion

As we expected, Afromontane forests are less sensitive to thermal changes than those in the Neotropics, as they are less diverse[22] and less dynamic[26] than Asian or Neotropical forests, but we still found that Afromontane forest tree communities are shifting towards more warm-affiliated species. Thermophilization in Afromontane tree communities was evident when considering all stems equally in a stem abundance-weighted metric (+$0.0045\,°C·y^{-1}$) but was weaker when calculated weighting for stem basal area (+$0.0020\,°C·y^{-1}$ and non-significant, cf. +$0.0066\,°C$ yr$^{-1}$ in the Andes[7]).

The greater signal for thermophilzation when considering changes in stem abundance than changes in basal area supports the role of recruitment and turnover of smaller stems, with a weaker signal for change in the basal area weighted metric due to the slower response of large stems. This contrast between basal-area weighted and stem abundance-weighted metrics is expected to be especially important in Afromontane forests, as they have longer carbon residence times than South American forests[23] due to the greater dominance of long-lived large trees[21]. This slower response could explain

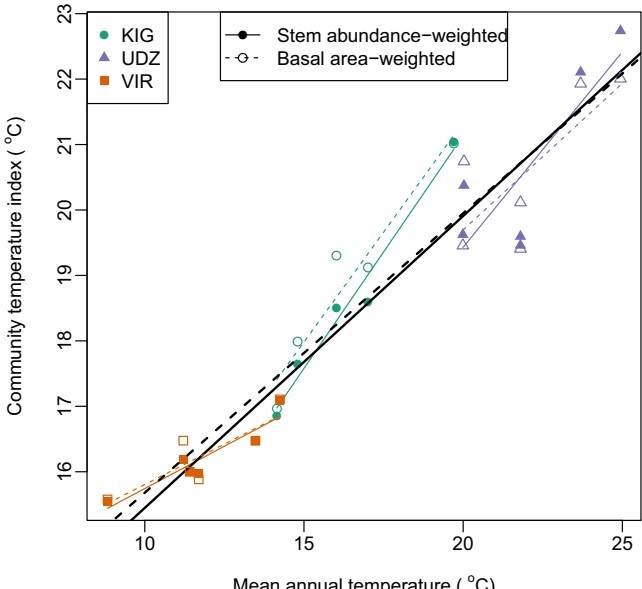

**Fig. 2 | Relationship between community temperature index (CTI) and mean annual temperature.** Coloured lines show regression relationships within each site, and the black line shows the regression relationship across all sites ($n = 17$ independent sampling plots). Both metrics of community temperature index were positively correlated with mean annual temperature (Spearman's rank correlation: $CTI_{Stem}$, $r_s = 0.945$, df = 15, $P < 0.001$, $CTI_{BA}$, $r_s = 0.942$, df = 15, $P < 0.001$), KIG Kigezi Highlands in Uganda, VIR Virunga Mountains in Rwanda-Uganda, UDZ Udzungwa Mountains in Tanzania. Source data are provided as a source data file.

why we observed weaker thermophilzation rates than in the Andes[8], despite having monitored a similar number of plots for a similar length of time. This expectation would only hold if larger trees were not experiencing widespread mortality under rising temperatures, and our finding of weaker thermophilization assessed with basal area weighting supports that this widespread mortality has not happened. Indeed, the largest stems (≥70 cm diameter) in our dataset tended to have warmer thermal optima than the smallest stems (10–30 cm diameter, Supplementary Fig. 4), being more likely to be pre-adapted to warmer conditions. This means that the weaker basal-area weighted response may not just be driven by slower dynamics of large trees, but it could also relate to their warmer thermal affiliation.

While the importance of turnover in smaller stems is consistent with our expectations that they would respond faster to warming temperatures than the large stems that are especially dominant in African forests, the marked role of recruitment is notable. This role of recruitment, which is more evident in the stem abundance-weighted metric where small recruits are not downweighed, contrasts with the dominance of mortality in driving thermophilization in the Andes (e.g. basal-area weighted thermophilization due to mortality = +0.0056 °C·y⁻¹, compared to −0.0005 °C·y⁻¹ due to recruitment[7]). Disturbances are known to accelerate thermophilization rates [e.g. ref. 11] and provide space for recruits to colonise. However, natural disturbance rates are expected to be lower in African forests than in the Andes as, for example, tropical cyclones are largely absent in mainland Africa (except in Mozambique[27]), lava flows are limited even in the active volcano of Mount Cameroon[28], and there are fewer areas with high landslide susceptibility in mountains in tropical Africa than in the Andes[29]. Another possibility is that thermophilization is being driven by recovery from past human disturbances. Differences in human disturbance are unclear; all plots sampled were considered old-growth and structurally intact (see Methods), but the human population density in the Kigezi Highlands and the Virunga Mountains is high

(300 and 590 people per km² respectively[30,31]) so we cannot discount past human disturbances. Evidence for recovery from past disturbance in our study sites is equivocal; while no systematic trends in community weighted mean wood density were seen (a trend towards species with higher wood density would be expected with succession), there was a shift in recruits towards more shade bearers (Supplementary Fig. 5). We consider it unlikely that variation in disturbance rate explains the general thermophilization trend or differences between Africa and the Andes, as (i) species' contributions to thermophilization trends were not associated with their wood density or light requirement (Supplementary Table 3), (ii) there was no relationship between thermophilization trends and turnover rate (Supplementary Table 4), indicating that the most dynamic plots were not changing fastest, and (iii) more frequent natural disturbances in the Andes should lead to random (with respect to temperature affiliation) mortality and non-random recruitment, yet a major difference Africa and the Andes is the greater role of non-random mortality in driving thermophilization in the latter.

There is a tendency for Afromontane tree species to have broader temperature niches than Andean tree species (see Supplementary Fig. 6), with some species having extensive lowland and montane distributions [e.g. ref. 32], and it is notable that we observe thermophilization in Afromontane forests despite this. It is possible that broad niched species (e.g. with distributions stretching from lowlands into mountains) could provide a large pool of potential recruits to colonise forests as temperatures increase. However, neither recruits nor trees that died shifted towards broader niched species (Supplementary Fig. 7). While it is possible that the greater niche breadth of Afromontane species slowed thermophilization rates, it did not prevent thermophilization, and the contrast between stem abundance and basal-area weighted metrics is more consistent with differences from the Andes being driven by differences in forest structure. A further difference in thermophilzation patterns between Afromontane and Andean forests is that we did not find any relationship between elevation and changes in CTI (Supplementary Table 4), which contrasts with the Andes were changes in CTI declined with elevation[7]. Elevation effects in the Andes have been attributed to the presence of specialised tree communities at the transitions between distinct habitats, such as at the timberline or at the base of the cloud forest[7]. Such specialised tree communities may be observed in some mountains in Africa, but not in all of them[33].

We had divergent expectations for how thermophilization would affect forest carbon function. If thermophilization was driven by tree mortality, then it could lead to carbon loss, whereas the recruitment of larger, warmer affiliated species could lead to carbon gains. Overall, during our study period plots acted as a carbon sink, with carbon gains through recruitment and growth exceeding carbon loss through mortality. However, this is unlikely to have been caused by thermophizlation, as carbon fluxes, stocks and changes in sink strength were unrelated to thermophilzation (Supplementary Table 4). Instead, a possible explanation is that the carbon sink (0.95 Mg ha⁻¹ yr⁻¹) represents the same response to $CO_2$ fertilisation observed in lowland African[23], South American[34] and Asian[35] tropical forests, and in Andean montane tropical forests[36]. However, despite the absence of any effect of thermophilization on carbon stocks at present, we suggest caution for the future. Potential interactions between increased temperatures and changing rainfall patterns, or edge effects should not be neglected, as Afromontane forests are highly threatened by deforestation[21].

In sum, we find that tree communities in Afromontane forests have shifted towards more warm-affiliated species, in alignment with findings from the Neotropics. While we find no evidence that shifts in species composition have affected carbon stocks, it will be important to determine whether continued thermophilization of Afromontane

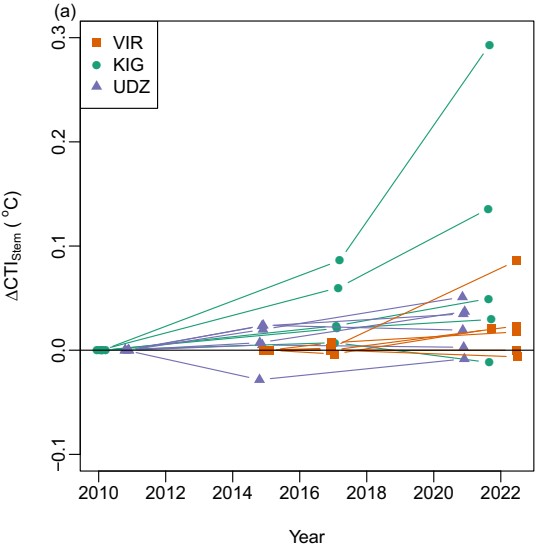
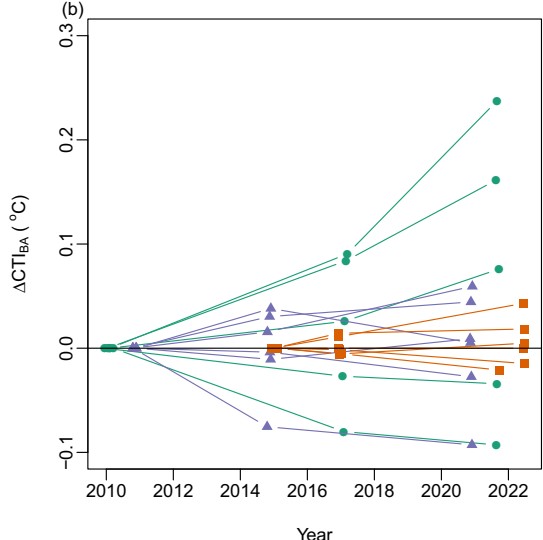

**Fig. 3 | Change in community temperature index (CTI) between censuses at each plot. a** Change in stem abundance-weighted CTI and **b** change in basal-area weighted CTI. Changes are expressed relative to values in the first census. KIG Kigezi Highlands in Uganda, VIR Virunga Mountains in Rwanda-Uganda, UDZ Udzungwa Mountains in Tanzania. $n = 17$ independent sampling plots. Source data are provided as a source data file.

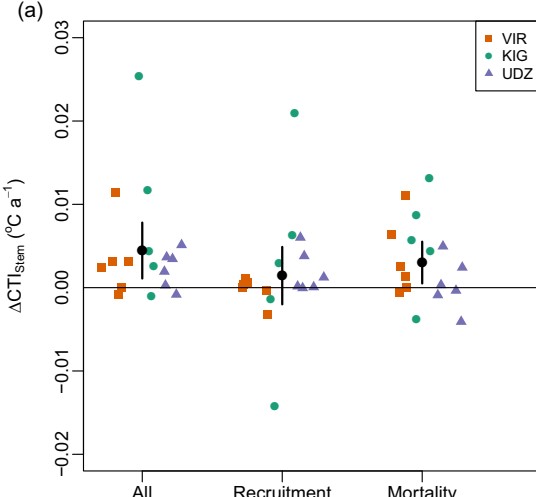
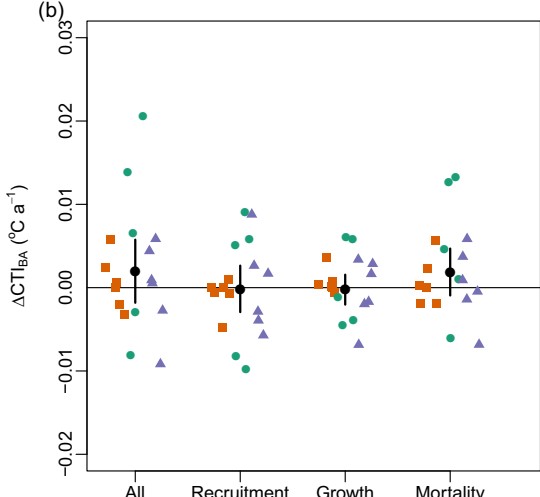

**Fig. 4 | Contribution of recruitment, mortality and growth to change in community temperature index (CTI). a** Change in stem abundance weighted CTI. **b** Change in basal-area weighed CTI. Points show changes for each plot, coloured by site. Black circles show average changes across plots, with error bars indicating 95% confidence intervals of changes ($n = 17$ independent sampling plots). CTI has been calculated using all species with at least 10 distribution records; see Supplementary Fig. 1 for consequences of using alternative thresholds for including species. Changes in CTI (all) have been decomposed into changes due to recruitment, changes due to mortality and (basal-area weighted only) change due to growth. KIG Kigezi Highlands in Uganda, VIR Virunga Mountains in Rwanda-Uganda, UDZ Udzungwa Mountains in Tanzania. Source data are provided as a source data file.

forests will increase or disrupt their ability to act as carbon stores and sinks in a warmer world.

## Methods

### Research permits

Uganda: a research permit was obtained from the Uganda Wildlife Authority (COD9605) and from the Uganda National council for Science and Technology (No NS282ES). Tanzania: a research permit was obtained from the Tanzania Commission for Sciences and Technology (No 2021-104-NA-2021-212). Rwanda: the Rwanda development Board (RDB) Tourism and Conservation granted permission to E.H.M. for carrying out this research in Volcanoes National Park in May 2022 (no specific permit number issued). No plant samples were collected and exported for this research.

### Inclusion & ethics

This research follows The TRUST Code of Conduct for Equitable Research Partnerships (https://www.globalcodeofconduct.org/). Specifically, this research: i) included local researchers throughout the research process, including authorship; ii) it was locally relevant, and findings were discussed with park managers; and iii) responsibilities were agreed ahead of research and capacity-building plans were discussed. The study was not approved by local ethics review committees as it did not involve research with humans or animals. We have taken

local and regional research relevant to our study into account in citations where available.

## Overview study areas

All study areas have a bimodal rainfall regime. Because of their different elevations and latitudes, forests on the Udzungwa Mountains (2576 m asl) tend to be hotter and drier than those in the Kigezi Highlands (2607 m asl) or the Virunga Mountains (4507 m asl) (see Fig. 2 for temperature ranges in study sites, and Supplementary Fig. 2 for precipitation).

## Plot data

We sampled 17 1-hectare inventory plots located between 610 and 3388 m asl, at three points in time between 2010 and 2022. These were initially part of the TEAM network[37] and are currently curated at www.ForestPlots.net[38]. Plots refer to old-growth and structurally intact (e.g. not affected by recent selective logging) closed-canopy evergreen wet or moist tropical forests. In all plots and census, tree diameter of all stems ≥10 cm diameter was measured at 1.3 m along the stem from the ground (or above buttresses if present) and each stem was identified to species. In the last census, tree height (measured using a handheld laser Nikon Forestry Pro) was also recorded for about 60 stems per plot comprising several individuals from each diameter class, as recommended by Sullivan et al.[39]. Palms, tree ferns and lianas were not sampled. Families and species names follow the African Plant Database (http://africanplantdatabase.ch). The final dataset consists of 9528 stems, of which 96.7% were identified to species, 98.9% to genus and 99.0% to family. Plot elevations were gathered with a GPS.

## CTI and CPI

Following previously established protocols (e.g. ref. 9), we estimated the thermal optima of all tree species that occurred in the study plots based on i) the locations of herbarium specimens reported for these species from tropical Africa according to the GBIF data portal (https://www.gbif.org/; data downloaded on 15 July 2022), and ii) the locations of these species in the AfriMont dataset, a dataset comprising 226 plots in 44 mountain regions in Africa (see ref. 21), excluding the three study sites to avoid circularity. GBIF records were inspected, and records that were tagged by the GBIF as having possible coordinate issues or that had obvious georeferencing errors (e.g., falling in large bodies of water) were discarded. Duplicate records from the same 30 arc-second grid cell were discarded. Confidence about species thermal optima is expected to increase with the number of occurrence records. To examine this, we subsampled records of each species and calculated the error in the species thermal optima compared to if all records had been used. This revealed an inflection at around 10 records (Supplementary Fig. 8), so this was used as a minimum threshold for including records. This resulted in a mean of 91.9 records ±85.2 SD per species. We explored the sensitivity of our results to using no threshold, or a higher threshold (30 records), and also of restricting records to those since 1980 so that occurrence records were temporally matched with climate data. The more stringent thresholds (limiting by date, restricting to 30+ records) had a limited effect on point estimates of change but did affect whether or not changes in CTI were statistically significant (Supplementary Fig. 1a). However, these more stringent thresholds reduced the proportion of stems for which species thermal optima were available (for example, going from 10 to 30 records as the threshold with no time cutoff meant going from 93.0% to 80.1% of stems being included, or from 166 to 128 species being included) and weakened the observed relationship between CTI and elevation (which we assume is related to the degree our estimates of CTI represent true environmental gradients). Therefore, the thresholds which removed the most species and resulted in the weaker characterisations of environmental relationships with CTI which resulted in non-significant patterns of change in CTI (Supplementary Fig. 1b).

The mean annual temperature at each location a species was observed at was extracted from ~1 km resolution from the Chelsa 1981–2010 climatology[24] and the thermal optima for each species were obtained by averaging the mean annual temperature at locations where the species was recorded. Our results are not affected by using climate data from Worldclim V2[40] instead of Chelsa (Supplementary Fig. 1a).

The community thermal index (CTI) for each plot and census was calculated in two ways. Firstly, we took the stem abundance weighted mean of species thermal optima (i.e., assigning a thermal optima to each stem, and then taking the mean across stems, $CTI_{Stem}$). Secondly, we calculated $CTI_{BA}$ as the thermal optima of the species weighted by their relative total basal area (summed cross-sectional stem area of all conspecifics measured at 1.3 m above ground or above buttresses if present in that plot and census). Changes in $CTI_{Stem}$ and $CTI_{BA}$ over time therefore integrate the effects of recruitment, mortality and, for $CTI_{BA}$ growth, on community composition.

These procedures were repeated for community precipitation index (CPI), using mean annual precipitation instead of mean annual temperature, also obtained from the Chelsa 1981–2010 climatology[24].

## Changes in CTI and CPI

We calculated change in $CTI_{Stem}$ and $CTI_{BA}$ between the last and first censuses and divided this change by the total census interval to obtain an annual rate of change. To test whether these trends differed from zero (i.e. whether there was a systematic trend towards thermophilization) we used a one-sample Wilcoxon test, and used bootstrap resampling (1000 random samples with replacement) to estimate 95% confidence intervals of changes. This approach was selected because examination of linear model residuals revealed that they were not normally distributed. However, we obtained similar results if we did use an intercept-only linear model (i.e. equivalent to a one-sample t-test); mean change in $CTI_{Stem} = +0.0045\,°C.y^{-1}$, 95% confidence intervals = 0.0011–0.0078, $t = 2.8$, $P = 0.013$. This effect remained positive and significant when any individual plot was removed (see Supplementary Table 5). These procedures were repeated for community precipitation index (CPI).

To assess the contribution of recruitment and mortality to changes in CTI, we performed an additive partition of changes into these effects. We calculated the CTI of the following groups of stems: trees present in the first census ($Alive_F$), trees present in the last census ($Alive_L$), trees in the first census that survived to the last census ($Survive_F$) and trees in the last census that had been alive since the first ($Survive_L$). Contributions due to mortality were calculated as $Survive_L$ - $Alive_F$, and contributions due to recruitment as $Alive_L$ – $Survive_F$.

## Historic temperature and precipitation changes for the study areas

Monthly temperature and precipitation data were obtained at 30 arc-second resolution for from the start of the study period (2010) until 2019 (most recent available data) from the CHELSA V2.1 timeseries[24]. We used linear models to relate annual precipitation and annual mean temperature at each plot location to year and report the slope of these relationships.

## Factors affecting CTI

We assessed whether the annual rate of change in CTI was related to plot elevation, stem density, percentage of stems identified to species, species richness and stem turnover rates using Spearman's rank correlation. To quantify the contribution of individual species to trends in CTI we followed[41] and used a jacknife approach where each species was removed from the dataset, CTI and change in CTI calculated without

the species, and the resultant trend compared to the original trend as contribution = $(\Delta CTI_{jackknife} - \Delta CTI_{all})/ \Delta CTI_{all}$. Here, negative contributions indicate species which are increasing the change in CTI (i.e. the trend in CTI is less positive without them), and positive contributions indicate species that are reducing change in CTI (i.e. the trend in CTI is more positive without them). We then related each species' contribution to change in CTI to i) species' wood density (obtained from ref. [42] and matched at the finest taxonomic resolution possible – 48.2% at family level, 25.2% at genus level, 24.9% at family level), ii) maximum size of the species (in terms of tree height, extracted from PROTA database https://www.prota4u.org/ or the African Plant Database https://africanplantdatabase.ch/, available for 99.4% of species), iii) species' light requirements (classified into 0: no data, 1: pioneer light demander, 2: non-pioneer light demander, 3: shade bearer; with data obtained from refs. [43–47] or PROTA database, available for 93.4% of species), iv) species' abundance (average number of stems in all three census), v) species' abundance change over time (difference between last and first census) and vi) species' thermal niche breadth, which was computed as the difference between the 95th and 5th percentile of the mean annual temperature in all the locations where species was recorded. We looked at relationships with these variables in a single model, fitting all subsets and performing model averaging based on AIC weights using the MuMIn R package[48], and also by looking at bivariate relationships. For this analysis species' contributions to CTI change were square-root transformed (transforming absolute values before reapplying the original sign) to meet model assumptions of normality and homoscedasticity of residuals.

We assessed whether niche breadth (calculated as above, i.e. difference between 95th and 5th percentile of mean annual temperatures at locations a species was recorded at) differed between our dataset and the Andean dataset used by Fadrique et al.[7]. We restricted both datasets to species with 10 or more records, calculated the niche breadth of each species in each continent, and used a Wilcoxon test to test whether niche breadth differed. Variation in niche breadth was visualised using probability density estimates produced using kernel estimators for bounded data[49] in the bde R package[50].

To examine whether species' thermal optima varied amongst size classes, we summarised the variation in thermal optima in stems of four size classes; 10–30 cm diameter, 30–50 cm, 50–70 cm, and >70 cm. These size classes match those used in Cuni-Sanchez et al.[21] which showed the dominance of the >70 cm size class in contributing to plot aboveground carbon stocks. This analysis was restricted to the first census to avoid duplication of stems. We tested if species' thermal optima differed between size classes using a mixed-effects model with size class as a fixed effect and plot identify as a random effect, implemented in the lme4 R package[51] and using the Satterthwaite approximation to obtain $P$ values (lmerTest R package[52]).

### Carbon stocks and dynamics

For each plot and census we calculated the carbon contained in the aboveground woody biomass (AGB) of each tree using allometric models based on diameter, height and wood density[53]. Wood density values were obtained from a global database[39] and matched at the finest taxonomic resolution possible using the BIOMASS R package[54]. Heights were estimated by constructing plot-level Weibull and log-log height-diameter models and selecting the best performing model to estimate the heights of unmeasured trees. We converted AGB to aboveground woody carbon (AGC), assuming that AGC (in MgC ha$^{-1}$) is 45.6% of AGB[55]. Tree-level carbon estimates were summed to provide estimates for the plot-level carbon stock in each census. Carbon gains due to woody productivity, carbon loss due to tree mortality, stem recruitment and stem mortality were estimated following[56], which accounts for biases in these metrics due to variation in census interval. For partitioning contributions to carbon fluxes due to recruitment, mortality and growth, we used an alternative census-interval

adjustment[57] which more directly estimates the contributions of unobserved recruits and dead trees. The difference between carbon gains and carbon loss indicates whether a plot was acting as a carbon sink. Calculations of carbon stocks and dynamics were performed using the BiomasaFP R package[58].

To assess whether change in CTI had consequences for ecosystem functioning, we assessed the Spearman's rank correlation between change in CTI and carbon stocks, gains and net change in carbon stocks. We also assessed the Spearman's rank correlation between each partitioned element of change in CTI (e.g. change due to recruits) and the corresponding carbon flux (i.e. carbon gains due to recruitment). We performed all analyses in R v.4.2.2[59].

### Reporting summary

Further information on research design is available in the Nature Portfolio Reporting Summary linked to this article.

## Data availability

Source data for figures are provided with this paper. Data supporting the findings of this study are available as a data package in Forest-Plots.net [https://doi.org/10.5521/forestplots.net/2024_2]. Climate data were obtained from Chelsa (https://chelsa-climate.org/) and Worldclim V2 (https://www.worldclim.com/version2). Species occurrence records were obtained from the GBIF data portal (https://www.gbif.org/), and trait data obtained from the African Plant Database (http://africanplantdatabase.ch) and the PROTA database (https://www.prota4u.org/). Input data underlying statistical analyses are available at [https://doi.org/10.5521/forestplots.net/2024_2]. Source data are provided with this paper.

## Code availability

The R code used in this study is available as a data package in ForestPlots.net [https://doi.org/10.5521/forestplots.net/2024_2].

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

## Acknowledgements

We thank the people of the many villages and local communities who welcomed our field teams and became our field assistants, without whose support the plot dataset would not have been possible. The last

census of the plots was funded by Marie Skłodowska-Curie Actions Global Fellowships (number 74356, A. C.-S.) and Ecologists in Africa Awards by the British Ecological Society (number EA20/1272, E.H.M.). Other funding supported this work, including Biodiversity Informatics Training Curriculum program (BITC2) grant from the JRS Biodiversity Foundation (E.H.M.), a ARC Future Fellowship (FT170100279, A.R.M.) and the Natural Environment Research Council (NE/W003872/1, M.J.P.S.). The Udzungwa plots were established by the University of York under the Valuing the Arc project with funding from the Leverhulme Trust, then remeasured as part of the TEAM Network, implemented by MUSE – Museo delle Scienze.

## Author contributions

A.C.-S. conceived the study, with help from K.J.F. A.C.-S., E.U. and E.H.M. led the field campaigns, with help from N.A.M., G.A.M., A.N., R.T and L.T. R.B., C.K., A.R.M., F.R. and D.S. contributed to facilitating field campaigns and were involved in data cleaning and analyses of initial censuses. A.C.-S., E.H.M. and A.S.K.N. gathered the GBIF data. A.C.-S., E.H.M. and M.J.P.S. analysed the plot data from all censuses and wrote the manuscript, with help from K.J.F. All co-authors read and approved the manuscript.

## Competing interests

The authors declare no competing interests.
