## [Peer Review File · Nature Communications]

Evidence of thermophilization in Afromontane forestsREVIEWER COMMENTS

Reviewer #1 (Remarks to the Author):

As the title indicates, Cuni-Sanchez et al present the first record of thermophilization in African montane forests. The main finding is novel and significant for ecology, global change and related fields. However, the authors have not enough support for their statement about thermophilization being a pantropical phenomenon. As they indicate, thermophilization has been recorded in many places in the Neotropics but not so much elsewhere.

There are two main concerns about the study.

The approach used to report thermophilization in the Neotropics (Duque et al 2015, PNAS 112,24; Fadrique et al 2018, Nature, 564; Feeley et al 2013, GCB, 19,11) is to calculate the rate of change in the community temperature index weighted by tree basal area of each forest plot used in the study. Basal area weighting is used as changes in basal area strongly influence forest structure and dynamics. However, as reported by the authors in lines L116-117 'Basal-area weighted CTI also increased over the study period ($0.0020\text{ }^{\circ}\text{C}\cdot\text{y}^{-1}$), but not significantly ($t = 1.1$, $df = 16$, $P = 0.286$)'. To be able to compare the results of this study with those of the Andes, basal-area weighted CTI is the metric that needs to be used (i.e. results from both regions need to be compared using the same metric), but this metric suggests that there is no thermophilization, which is problematic.

The thermophilization results are shown in Figure 3. The results for all plots and for recruitment only could be biased by the outliers (one above $0.02\text{ }^{\circ}\text{C}/\text{yr}$) in 'All' and two outliers in 'Recruitment only' (one around $0.02\text{ }^{\circ}\text{C}/\text{yr}$ and below $-0.01\text{ }^{\circ}\text{C}/\text{yr}$). The analysis should also be presented without the outliers and evaluate whether the thermophilization results and claims about recruitment persist.

Further comments

-Thermal migration rate is introduced at the start of the manuscript (L32-L42), however the results are presented in terms of change in CTI, although this is the same it is clearer to have

a consistent terminology in the whole the text. The y-axis of figure 3 is delta CTI. The unifying term used may be thermophilization rather than thermal migration rate (or delta CTI) which assumes that there is migration.

-L90 -L91 The sentence could be modified for ease of understanding. Also, to explain the sentence better, the authors should also include respective mean annual temperatures at each site as a function of elevation to see whether CTI does track environmental lapse rate.

-L102-L104 The authors should comment on the goodness of the data set used to estimate average warming rates for each site but also should include a figure of the thermophilization rate and rate of warming at each site to show how warming in causing thermophilization over time.

-The methodology needs to mention the diameter size class used in the study. The recruitment component is highly affected by the minimum diameter class included in the study. If the reported results are based on trees with diameter at breast height (DBH) lower than 10 cm, the analyses can be repeated only using trees larger than 10 cm of DBH to be comparable to Neotropical estimates (Fadrique et al 2018, Nature, 564) and only for trees lower than 10cm (i.e., juveniles) and to compare to Duque et al. 2015, PNAS 112,24.

-The carbon stocks results do not currently add much to the main message of the paper. However, what would be most useful is to present is the carbon contribution from recruitment, growth, and mortality separately and discuss if the recruitment of new thermophilic species modifies total carbon stocks.

L145 -L149

‘Although our primary result is an overall similarity of thermophilization, we also found some subtle differences ‘

This sentence needs to be clearer, similarity to what? It becomes clear in the next sentence but it should be clear from the start.

‘Firstly, the average TMR we report from Africa (+0.0045 °C·y⁻¹) is weaker than rates reported from the Neotropics ([5], although they are just within the 95% CI of our estimate), and unlike in the Neotropics, we find no significant change in basal-area weighted community thermal associations. ‘

The value for the Neotropics needs to be stated in the above sentence, but the comparison has to be done using with the same methodology otherwise the results are not comparable.

L149-L152, what are the values for the Andes (all, recruitment, and mortality, again include values using comparable methodologies) This will allow the reader to understand the comparison better.

158-160 Unclear why recruitment is more important, relative to the Andes? What are those values?

The introduction (L51-L69) includes comparison of Neotropical and African forest (but nothing on disturbance) and it includes a hypothesis regarding broad ecological niches in African forest being less sensitive to climatic changes. Although the discussion (L158 -162) touches upon the ecological niches and mentions that Afromontane forests species have broad niches, the hypothesis mentioned in the introduction is never tested. The authors could show a comparison of thermal (and precipitation) niche amplitudes for the Andes and for their data set to evaluate the hypothesis they state. The discussion text from L163-L176 focuses on comparing disturbance in Africa and the Andes, however this topic was not identified in the introduction.

L58 replace ‘orthographic’ with ‘orographic’.

Legend of Extended figure 3—replace ‘breath’ with ‘breadth’.

Reviewer #2 (Remarks to the Author):

The authors report thermophilization of plant communities in Afromontane forest systems based on re-census data in 17 forest plots. They unravel the relative contribution of

recruitment versus mortality to trends in thermophilization. Furthermore, the aim to link thermophilization to changes in carbon storage.

I believe that the analyses on the relative contribution of recruitment versus mortality are interesting and a novel aspect of this manuscript. However, (1) I do miss some clear hypotheses that logically follow from the introduction; and (2) several methodological decisions hamper the interpretation of the results.

Below, I provided numerous comments line-by-line. I hope you find them useful to further improve your work. Note that you can consider these comments as suggestions, although I believe that some critical info is missing or not always clear from the text.

-title and abstract: can 'forests' be replaced by 'tree species'? This is a suggestion, but I think it is important since I found it not clear (throughout the whole text) on which species group you focus. Tree species, or were understorey herbs, lianas etc. also included?

-L12: Lower tree diversity compared to what? From the text at L54, I assume that you are referring to the Neotropics? Please clarify. It is a bit confusing since you also refer to temperate regions at L10

-L23-24: this trend is also observed based on empirical data. I would rephrase this sentence to make clear that this is not only a predicted trend, but also a real observation. (See also meta-analyses from e.g. Lenoir 2020; Chen 2011).

-L26: unclear what you mean here with 'at any given location'. Please remove or clarify

-Paragraph starting at L32: I am wondering if it is necessary to present the different case studies in such detail. It breaks the storyline of the introduction. My suggestion would be to summarize this paragraph and integrate this into the next one.

L58-59. Perhaps good to back up this statement with a reference.

L60-62: To keep the intro more general, I would not mention the example of one specific species. I assume that you can find such examples of generalist species almost everywhere.

L71: suggest to replace 'forests' by 'tree species'

L77: The carbon stock story comes a bit out of the blue here. It would help if you could link a clear hypothesis to it. I.e. why would you expect carbon stocks to change, and in which direction? Similarly, it would be strong if you could link hypotheses to the tests explained at L73 (thermophilization) and at L74 (relative contribution of tree mortality and recruitment). For thermophilization, do you expect low/no rates of thermophilization since Afrotropical

forests are less sensitive to increased temperatures (cf. txt on L66). For the demographic processes, do you expect mortality of recruitment to be the most important process (cf. text on 43-48)? For changes carbon stocks, I do not find information in the introduction.

L81: It would be good if you could here inform the reader about the temperature range across the studied plots.

L82: What was the average number of species observed per plot?

L82-L85: OK, but on how many individuals and species was this based. Unless I missed it, I did not find the number of species that were recorded across the whole plot network. A small introducing paragraph with general information about the data set, similar like the first paragraph of the results section in Duque et al, 2015 PNAS

(<https://www.ncbi.nlm.nih.gov/pmc/articles/PMC4553780/>) would help here a lot.

-L84-90: Are these results from the parametric regression? Since the CTI data was non-normally distributed, I would include here some cautions.

L97-98: same question as the previous. Perhaps a reference to the results of the non-parametric tests would help here. Also a reference to Extended Data Figure 6 is necessary to add nuance to the results, especially since significance depended on the threshold value.

L141: So this contradicts your hypotheses that Afromontane forests are less sensitive to thermal changes (cf. L66-68)? Then I would explicitly mention this

L161: Do you mean thermal niches? Please specify

L163: I am a bit confused here. Weren't the analyses based at the community-level?

L198: was this a significant correlation or not? Please clarify.

Fig 3: I suggest to add the reference to ED Fig6 here. I think this is important since the significance of the trends were dependent on methodological decisions.

L 342-352. Perhaps it is interesting for the readers to also have some information about the temperature and precipitation ranges observed in the plots? E.g. an average over the studied period based on the CRU-TS data set.

L352: OK, but how many species in total?

L357: It is common practice to also provide to the DOI download link from GBIF.

L363: It would be interesting to also provide the mean number of records per species (after data cleaning).

L365: Was this based on visual inspection?

L370: could you provide the proportion of species for which species thermal optima could

be inferred for year threshold? Was there a lot of defence between 10 and 30+ records?

L378: Typo?

L379: Chelsa climatology is an average from 1981-2010 (not 1980).

L413: Why 1980 was chosen as a baseline? Isn't it more logical to consider the temperature and precipitation change within the time interval of the study period?

L434: I suggest to consider using the 5 and 95 percentiles of the species' temperature distributions to quantify thermal niche breadth. This avoids extreme wide thermal ranges due to species' records in marginal environmental conditions.

L440: typo: homoscedasticity

Reviewer #3 (Remarks to the Author):

This reviewer has also provided a marked-up version of your manuscript, please see attached file

The paper of Cuni-Sanchez et al. documented changes in species composition in 17 1-ha plots from three locations in the tropical mountains of Africa, leading to changes in community thermal scores, defined as thermophilization. The analyses presented added novel elements to understanding CTI changes in the context of forest tree dynamics and provided new insights into understanding climate change impacts on tropical mountain ecosystems. Plus, the authors sit on a very nice and unique dataset. Likewise, the paper is very well written, and the analyses are sound. As such, it would be well suited for eventual publication in this journal. However, in my opinion, the manuscript is too premature for publication in its current version. A few concerns demand careful attention, particularly within the methods and results sections.

Below, I resume my main comments and provide more details in the attached file.

1. Using a 1-km² pixel size of the climatic variables to derive the thermal optima and "precipitation" optima of the species in tropical mountains is problematic. This rather coarse grain could lead to averaging temperature and precipitation values that are quite different, especially in ecotonal boundaries. I suggest performing a downscaling of the climatic grids using the SRTM as previously documented (See:

<https://doi.org/10.1111/jbi.13759> and <https://doi.org/10.1038/s41586-018-0715-9>) and recalculate the analyses. This could provide more accurate estimates and provide more insights into the observed trends.

2. The high uncertainty in the annual and monthly precipitation grids for tropical mountain regions, plus the no significant trends between CPI and annual precipitation values at plot locations nor with precipitation changes or CTI changes over time, suggests that this part of the analyses could be better placed in the supplementary material. In this way, the two approaches of thermophilization analyses (abundance vs basal area) and the factors related to the observed CTIs could be presented in detail. In fact, most of the Discussion section has almost no mention of the CPI results. Expanding the implications of recruitment vs mortality rates on the CTIs could be more relevant in the context of the manuscript.

3. Values reported in results for changes in CTI do not match those presented in Supplementary table 1. Perhaps the values presented in the main body of the manuscript correspond to bootstrapped means, whereas the values in Supplementary table 1 are just the observed values. In any case, both sections should contain the same values to convey the same message throughout the document. In addition, this table should be part of the main text, given the importance of the information presented.

4. In the introduction section, the authors refer to the profound transformation of the landscape in the African highlands during the Pleistocene climatic changes, leading to forest contraction that prevailed only in small scattered refugia or, in extreme cases, forest disappeared utterly to be later colonized. These documented changes shaped the current distribution of individual species. Thus, the lower thermophilization rates observed in the African highlands compared to those documented in SA could be in part explained by these past changes and the current “relictual” distribution of the African tree species. It would be interesting if the authors could elaborate more on this subject and the potential implications of their findings in the discussion section.

Reviewer #1 (Remarks to the Author):

As the title indicates, Cuni-Sanchez et al present the first record of thermophilization in African montane forests. The main finding is novel and significant for ecology, global change and related fields. However, the authors have not enough support for their statement about thermophilization being a pantropical phenomenon. As they indicate, thermophilization has been recorded in many places in the Neotropics but not so much elsewhere.

There are two main concerns about the study.

The approach used to report thermophilization in the Neotropics (Duque et al 2015, PNAS 112,24; Fadrique et al 2018, Nature, 564; Feeley et al 2013, GCB, 19,11) is to calculate the rate of change in the community temperature index weighted by tree basal area of each forest plot used in the study. Basal area weighting is used as changes in basal area strongly influence forest structure and dynamics. However, as reported by the authors in lines L116-117 'Basal-area weighted CTI also increased over the study period (0.0020 °C·y⁻¹), but not significantly (t = 1.1, df = 16, P = 0.286)'. To be able to compare the results of this study with those of the Andes, basal-area weighted CTI is the metric that needs to be used (i.e. results from both regions need to be compared using the same metric), but this metric suggests that there is no thermophilization, which is problematic.

RESPONSE: Both abundance and basal-area weighted metrics were presented and differences were discussed. Far from being problematic, we can learn more about changes by comparing the different metrics. The abundance-weighted metric is more sensitive to changes in small stems (recruitment, mortality of smaller trees), while the basal-area weighted metric is more sensitive to the mortality of large stems. Finding a significant change in the abundance weighted metric but not the basal-area weighted one indicates that it is the smaller stems that are driving the pattern. We agree that making more of this comparison is beneficial, so have changed figures 2-4 so they now present both the abundance and basal-area weighted CTI, and restructured the results to give more prominence to the comparison of both metrics. We also note the reviewers concern that quantitative comparisons with the Andes should be based on the same metric, and we have amended our wording in the manuscript to ensure the comparison is clear.

The thermophilization results are shown in Figure 3. The results for all plots and for recruitment only could be biased by the outliers (one above 0.02 °C/yr) in 'All' and two outliers in 'Recruitment only' (one around 0.02 °C/yr and below -0.01 °C/yr). The analysis should also be presented without the outliers and evaluate whether the thermophilization results and claims about recruitment persist.

RESPONSE: We have conducted two sensitivity checks. Firstly, repeating the analysis with non-parametric statistics, which by using ranks are less sensitive to outliers, gives the same inferences regarding statistical significance (these are now presented in the manuscript following R2). Secondly, we checked the consequences of removing each plot from the analysis. For overall thermophilization, trends remained positive and significant when any individual plot was removed:

Plot removed	Slope	P
BWI-01	0.0048	0.011
VRU-04	0.0048	0.011
UDZ-01	0.0048	0.011
VRU-02	0.0047	0.012
UDZ-04	0.0047	0.013
UDZ-05	0.0046	0.015
VRU-06	0.0046	0.016

BWI-04	0.0046	0.016
VRU-01	0.0045	0.017
VRU-05	0.0045	0.017
UDZ-03	0.0045	0.017
UDZ-02	0.0045	0.017
BWI-03	0.0045	0.018
UDZ-06	0.0044	0.019
VRU-03	0.0040	0.026
BWI-02	0.0040	0.026
BWI-05	0.0031	0.005

For recruitment, we observed a non-significant positive effect with all the plots included. Changes remained positive but non-significant when any single plot was removed (0.0002 – 0.0024). We note that our conclusions about the role of recruitment are also based on comparing the stem-based and basal-area weighted results.

Overall these checks demonstrate that our results are not driven by single outlying plots.

Further comments

-Thermal migration rate is introduced at the start of the manuscript (L32-L42), however the results are presented in terms of change in CTI, although this is the same it is clearer to have a consistent terminology in the whole the text. The y-axis of figure 3 is delta CTI. The unifying term used may be thermophilization rather than thermal migration rate (or delta CTI) which assumes that there is migration.

RESPONSE: We now use the unifying term thermophilization in the introduction, and CTI (either CTI-A, CTI-B to refer to abundance or basal area weighted, respectively) in the rest of the manuscript, as these two metrics are the ones calculated, displayed in figures and discussed after.

-L90 -L91 The sentence could be modified for ease of understanding. Also, to explain the sentence better, the authors should also include respective mean annual temperatures at each site as a function of elevation to see whether CTI does track environmental lapse rate.

RESPONSE: This sentence has been modified. We now present CTI against temperature rather than elevation in Fig. 2 so it is easier to see how they track environmental lapse (despite a strong positive relationship, CTI changes by < 0.5C for each 1°C increase in temperature).

-L102-L104 The authors should comment on the goodness of the dataset used to estimate average warming rates for each site but also should include a figure of the thermophilization rate and rate of warming at each site to show how warming in causing thermophilization over time.

RESPONSE: In response to R2 we now look at change over the study period rather than over a longer time period, and use Chelsa timeseries rather than CRU. The Chelsa monthly climatologies are based on ERA reanalysis of numerical models so include modelled lapse rates (which will vary between air masses) and although there are limitations (e.g. behaviour during temperature inversions, resolution of numerical models for picking up micro-climate) they probably represent the best available data without having in situ meteorological stations. Even so, we don't think it would be valid to look at trends *within* sites, so just present the site-level mean trends. The rank order of thermophilization rates matches the rank order of warming rates, although with three sites we can't formally analyse this relationship.

-The methodology needs to mention the diameter size class used in the study. The recruitment component is highly affected by the minimum diameter class included in the study. If the reported results are based on trees with diameter at breast height (DBH) lower than 10 cm, the analyses can be repeated only using trees larger than 10 cm of DBH to be comparable to Neotropical estimates (Fadrique et al 2018, Nature, 564) and only for trees lower than 10cm (i.e., juveniles) and to compare to Duque et al. 2015, PNAS 112,24.

RESPONSE: Our analysis is based on stems with diameter at breast height (or above buttress) of 10 cm or greater. We now state this in the main text. We did not measure smaller stems and cannot compare juveniles vs adult trees like in Duque et al. 2015 paper.

-The carbon stocks results do not currently add much to the main message of the paper. However, what would be most useful is to present is the carbon contribution from recruitment, growth, and mortality separately and discuss if the recruitment of new thermophilic species modifies total carbon stocks.

RESPONSE: We agree, this is a good suggestion, and now present contributions to carbon dynamics from recruitment, growth and mortality (see Supplementary Table 1), and relate these to thermophilization partitioned into each component. We also added a new analysis in which we investigated if large trees (defined as >70cm diameter) are more thermophilic or not than smaller size species (see new figure Extended Data Fig. 4), and discuss the implications of such findings for future carbon stocks in the discussion.

L145 -L149

‘Although our primary result is an overall similarity of thermophilization, we also found some subtle differences ‘

This sentence needs to be clearer, similarity to what? It becomes clear in the next sentence but it should be clear from the start.

RESPONSE: This paragraph has been reworded: see lines 158-164.

‘Firstly, the average TMR we report from Africa (+0.0045 °C·y⁻¹) is weaker than rates reported from the Neotropics ([5], although they are just within the 95% CI of our estimate), and unlike in the Neotropics, we find no significant change in basal-area weighted community thermal associations. ‘
The value for the Neotropics needs to be stated in the above sentence, but the comparison has to be done using with the same methodology otherwise the results are not comparable.

RESPONSE: We are now clear that the Andes value is basal-area weighted, and report our basal-area weighted value as well here.

L149-L152, what are the values for the Andes (all, recruitment, and mortality, again include values using comparable methodologies) This will allow the reader to understand the comparison better.

RESPONSE: We have added these values, although they are restricted to basal-area weighted values as these are the only ones that could be extracted by digitising graphs.

158-160 Unclear why recruitment is more important, relative to the Andes? What are those values?

RESPONSE: In the Andes recruitment had a weak negative effect on thermophilization (ref 7), compared to the non-significant but positive effect we find here. Values now reported.

The introduction (L51-L69) includes comparison of Neotropical and African forest (but nothing on disturbance) and it includes a hypothesis regarding broad ecological niches in African forest being less sensitive to climatic changes. Although the discussion (L158 -162) touches upon the ecological niches and mentions that Afromontane forests species have broad niches, the hypothesis mentioned in the introduction is never tested. The **authors could show a comparison of thermal (and precipitation) niche amplitudes for the Andes and for their data set** to evaluate the hypothesis they state. The discussion text from L163-L176 focuses on comparing disturbance in Africa and the Andes, however this topic was not identified in the introduction.

RESPONSE: We have added an analysis based on GBIF records used in Fadrique et al. (2018), which indicates that while broad-niched species are found in both the Andes and Africa, there is a greater proportion of narrow-niched species in the Andes. See Extended Data Fig. 6.

L58 replace 'orthographic' with 'orographic'.

RESPONSE: Done.

Legend of Extended figure 3—replace 'breath' with 'breadth'.

RESPONSE: Done.

Reviewer #2 (Remarks to the Author):

The authors report thermophilization of plant communities in Afromontane forest systems based on re-census data in 17 forest plots. They unravel the relative contribution of recruitment versus mortality to trends in thermophilization. Furthermore, the aim to link thermophilization to changes in carbon storage.

I believe that the analyses on the relative contribution of recruitment versus mortality are interesting and a novel aspect of this manuscript. However, (1) I do miss some clear hypotheses the logically follow from the introduction; and (2) several methodological decisions hamper the interpretation of the results.

Below, I provided numerous comments line-by-line. I hope you find them useful to further improve your work. Note that you can consider these comments as suggestions, although I believe that some critical info is missing or not always clear from the text.

-title and abstract: can 'forests' be replaced by 'tree species'? This is a suggestion, but I think it is important since I found it not clear (throughout the whole text) on which species group you focus. Tree species, or were understorey herbs, lianas etc. also included?

RESPONSE: We thank the reviewer for raising this point, we only studied trees. We think it is better to keep forests in the title because tree species can be found in other habitats (e.g. savannas) but we clarified in abstract that we only focused on trees.

-L12: Lower tree diversity compared to what? From the text at L54, I assume that you are referring to the Neotropics? Please clarify. It is a bit confusing since you also refer to temperate regions at L10

RESPONSE: This relates to the Neotropics and other tropical forest regions. Now clarified.

-L23-24: this trend is also observed based on empirical data. I would rephrase this sentence to make clear that this is not only a predicted trend, but also a real observation. (See also meta-analyses from e.g. Lenoir 2020; Chen 2011).

RESPONSE: Changed to predicted and observed, suggested references added.

-L26: unclear what you mean here with 'at any given location'. Please remove or clarify

RESPONSE: Changed to relative abundance in communities – we are referring to the idea that range shifts will lead to local communities (e.g. trees in a forest plot) shifting towards more warm adapted species.

-Paragraph starting at L32: I am wondering if it is necessary to present the different case studies in such detail. It breads to storyline of the introduction. My suggestion would be to summarize this paragraph and integrate this into the next one.

RESPONSE: we reduced the text as suggested and merged with the next paragraph.

L58-59. Perhaps good to back up this statement with a reference.

RESPONSE: This statement already has a reference, we are unsure if the reviewer confused the line.

L60-62: To keep the intro more general, I would not mention the example of one specific species. I assume that you can find such examples of generalist species almost everywhere.

RESPONSE: Example removed, as suggested.

L71: suggest to replace 'forests' by 'tree species'

RESPONSE: Changed to tree communities in tropical montane forests

L77: The carbon stock story come a bit out of the blue here. It would help if you could link a clear hypothesis to it. I.e. why would you expect carbon stocks to change, and in which direction? Similarly, It would be strong if you could link hypotheses to the tests explained at L73 (thermophilization) and at L74 (relative contribution of tree mortality and recruitment). For thermophilization, do you expect low/no rates of thermophilization since Afromontane forests are less sensitive to increased temperatures (cf. txt on L66). For the demographic processes, do you expect mortality of recruitment to be the most important process (cf. text on 43-48)? For changes carbon stocks, I do not find information in the introduction.

RESPONSE: We added a new paragraph stating our hypothesis for both recruitment/mortality and carbon stocks, see lines 61-74.

L81: It would be good if you could here inform the reader about the temperature range across the studied plots.

RESPONSE: Mean annual temperatures are now shown for each plot in Fig. 2.

L82: What was the average number of species observed per plot?

RESPONSE: 23.4 species per plot. This is now added to the text.

L82-L85: OK, but on how many individuals and species was this based. Unless I missed it, I did not find the number of species that were recorded across the whole plot network. A small introducing paragraph with general information about the data set, similar like the first paragraph of the results section in Duque et al, 2015 PNAS (<https://www.ncbi.nlm.nih.gov/pmc/articles/PMC4553780/>) would help here a lot.

RESPONSE: We have added the following text giving the number of species and individuals.

New text: "This resulted in a dataset of 26,850 measurements of 9,528 stems belonging to 207 species (mean 23.4 species per plot), of which 166 species (93.0% of stems) had sufficient occurrence records to calculate their thermal optima" Lines 88-90.

-L84-90: Are these results from the parametric regression? Since the CTI data was non-normally distributed, I would include here some cautions.

RESPONSE: We originally decided to present results from parametric regression as we wish to make inferences about the rate of CTI change and its uncertainty. However, we note the concern and did check that our results held using non-parametric methods. While this was originally presented in the methods, we have now moved this to the main text, with our original linear model analysis moved to the SI methods.

L97-98: same question as the previous. Perhaps a reference to the results of the non-parametric tests would help here. Also a reference to Extended Data Figure 6 is necessary to add nuance to the results, especially since significance depended on the threshold value.

RESPONSE: We have now reversed the results so put the non-parametric results here and the parametric results are in the methods. We have also added a reference to ED Fig. 6 (which is now ED Fig. 1). (see lines 108-110)

L141: So this contradicts your hypotheses that Afromontane forests are less sensitive to thermal changes (cf. L66-68)? Then I would explicitly mention this

RESPONSE: sentence modified as suggested.

L161: Do you mean thermal niches? Please specify

RESPONSE: yes, sentence modified as suggested.

L163: I am a bit confused here. Weren't the analyses based at the community-level?

RESPONSE: yes, we focused on community-level, clarified.

L198: was this a significant correlation or not? Please clarify.

RESPONSE: not significant, sentence modified after adding new analysis.

Fig 3: I suggest to add the reference to ED Fig6 here. I think this is important since the significance of the trends were dependent on methodological decisions.

RESPONSE: The figure legend has now been modified to refer to the ED figure 6 (which is now ED Fig. 1).

L 342-352. Perhaps it is interesting for the readers to also have some information about the temperature and precipitation ranges observed in the plots? E.g. an average over the studied period based on the CRU-TS data set.

RESPONSE: We have modified Fig. 2 so the X-axis is now mean annual temperature, with precipitation ranges shown in a new Extended Data Fig. 2.

L352: OK, but how many species in total?

RESPONSE: We recorded 207 species in total, with 166 having sufficient records to include (93% of stems). This has now been added to the text (see our response to L82-L85 comment).

L357: It is common practice to also provide to the DOI download link from GBIF.

RESPONSE: We downloaded each species separately and for some species we had to download the records using both the 'old accepted' name and the new one. As we have over 200 species, it is not possible to provide all DOI here.

L363: It would be interesting to also provide the mean number of records per species (after data cleaning).

RESPONSE: Now added: "This resulted in a mean of 91.9 record \pm 85.2 SD per species."

L365: Was this based on visual inspection?

RESPONSE: yes.

L370: could you provide the proportion of species for which species thermal optima could be inferred for year threshold? Was there a lot of defence between 10 and 30+ records?

RESPONSE: Now added: "However, these more stringent thresholds reduced the proportion of stems for which species thermal optima were available (for example, going from 10 to 30 records as the threshold with no time cutoff meant going from 93.0% to 80.1% of stems being included, or from 166 to 128 species being included) and weakened the observed relationship between CTI and elevation".

L378: Typo?

RESPONSE: yes. Corrected.

L379: Chelsa climatology is an average from 1981-2010 (not 1980).

RESPONSE: corrected.

L413: Why 1980 was chosen as a baseline? Isn't it more logical to consider the temperature and precipitation change within the time interval of the study period?

RESPONSE: For temperature trends, we now look at change over the study period as suggested. However, we still use the 30-year climate mean for calculating species' thermal optima as this captures the typical long-term conditions at a location (shorter periods might be dominated by year-to-year variability rather than capturing spatial variation in climate).

L434: I suggest to consider using the 5 and 95 percentiles of the species' temperature distributions to quantify thermal niche breadth. This avoids extreme wide thermal ranges due to species' records in marginal environmental conditions.

RESPONSE: Changed as suggested.

L440: typo: homoscedasticity

RESPONSE: corrected.

Reviewer #3 (Remarks to the Author):

This reviewer has also provided a marked-up version of your manuscript, please see attached file

The paper of Cuni-Sanchez et al. documented changes in species composition in 17 1-ha plots from three locations in the tropical mountains of Africa, leading to changes in community thermal scores, defined as thermophilization. The analyses presented added novel elements to understanding CTI changes in the context of forest tree dynamics and provided new insights into understanding climate change impacts on tropical mountain ecosystems. Plus, the authors sit on a very nice and unique dataset. Likewise, the paper is very well written, and the analyses are sound. As such, it would be well suited for eventual publication in this journal. However, in my opinion, the manuscript is too premature for publication in its current version. A few concerns demand careful attention, particularly within the methods and results sections.

Below, I resume my main comments and provide more details in the attached file.

1. Using a 1-km² pixel size of the climatic variables to derive the thermal optima and "precipitation" optima of the species in tropical mountains is problematic. This rather coarse grain could lead to averaging temperature and precipitation values that are quite different, especially in ecotonal boundaries. I suggest performing a downscaling of the climatic grids using the SRTM as previously documented (See: <https://doi.org/10.1111/jbi.13759> and <https://doi.org/10.1038/s41586-018-0715-9>) and recalculate the analyses. This could provide more accurate estimates and provide more insights into the observed trends.

RESPONSE: We were cautious about this approach and the potential for false precision in biodiversity records, and also note that many species occur in lowland areas as well where these issues are not so important. However, we have assessed the effect downscaling would have on our results. Using SRTM elevation data to downscale temperatures at locations species were recorded at provides very similar results to our original analysis (Change in CTI = 0.0042, 95% CI = 0.0010 – 0.0073, t=2.8, P = 0.121). Because the results are similar, and our concerns with downscaling, we have retained our original approach working with 1km resolution climate data. However, we would happily present the downscaled analysis in the supporting materials if the reviewer thinks it important.

2. The high uncertainty in the annual and monthly precipitation grids for tropical mountain regions, plus the no significant trends between CPI and annual precipitation values at plot locations nor with precipitation changes or CTI changes over time, suggests that this part of the analyses could be better placed in the supplementary material. In this way, the two approaches of thermophilization analyses (abundance vs basal area) and the factors related to the observed CTIs could be presented in detail. In fact, most of the Discussion section has almost no mention of the CPI results. Expanding the implications of recruitment vs mortality rates on the CTIs could be more relevant in the context of the manuscript.

RESPONSE: We thank the reviewer for this suggestion, we have moved CPI results to supplementary material, and focus on the two approaches to thermophilization in the main paper. We also modified Figures 2, 3 and 4 to display both CTI metrics.

3. Values reported in results for changes in CTI do not match those presented in Supplementary table 1. Perhaps the values presented in the main body of the manuscript correspond to bootstrapped means, whereas the values in Supplementary table 1 are just the observed values. In any case, both sections should contain the same values to convey the same message throughout the document. In addition, this table should be part of the main text, given the importance of the information presented.

RESPONSE: We apologise for the discrepancy. The values in the table came from an earlier version of the analysis using Worldclim rather than CHELSA as the climate data. This has now been corrected.

4. In the introduction section, the authors refer to the profound transformation of the landscape in the African highlands during the Pleistocene climatic changes, leading to forest contraction that prevailed only in small scattered refugia or, in extreme cases, forest disappeared utterly to be later colonized. These documented changes shaped the current distribution of individual species. Thus, the lower thermophilization rates observed in the African highlands compared to those documented in SA could be in part explained by these past changes and the current “relictual” distribution of the African tree species. It would be interesting if the authors could elaborate more on this subject and the potential implications of their findings in the discussion section.

RESPONSE: We studied over 200 species, and while some are currently restricted to mountain ranges and could be considered that they have a ‘current relictual’ distribution’ (compared to their distribution in the past), others are currently still found in the lowlands and have relatively large thermal niches (see Extended Data Fig. 6). We thank the reviewer for this suggestion, but we have already modified the discussion considerably following other comments and we decided not to discuss these ‘relictual distribution’ extensively in the discussion.

Detailed comments:

L14 - seems a low number to draw conclusions for an entire Biome.

RESPONSE: we have plots in 3 different mountains in 3 different countries.

L17 - But mortality was equally important:Lines 112-114

RESPONSE: Changed to “both recruitment and mortality”

L25 - Add a tropical reference to enrich the European reference: <https://doi.org/10.1111/geb.13721>.

RESPONSE: Changed.

L57 - please, specify when in number of years.

RESPONSE: This depends on the mountain and the species, it is difficult to give a precise number here without adding extensive text sorry.

L59 - please provide more information on the similarities and differences in climates among the three studied mountain ranges.

RESPONSE: This has now been added in the section online methods, see lines 392-395.

L61 - Can you provide an overall mean of afro-montane species climatic niche breadth besides the specific example of *Prunus africana*? Does the referred precipitation gradient also apply to thermal conditions?

RESPONSE: The example of *Prunus africana* has been removed following suggestion by another reviewer.

L64 - I don't really follow why afro-montane species could be less sensitive than neotropical montane species

RESPONSE: because many of the tree species now found in Afro-montane forests are also found at lower elevations and colonised the mountains after forest contractions during the Pleistocene. We assessed this further by testing if the niche breadth of Afro-montane and Andean species differed, and found that on average Afro-montane species did, as expected, have broader niches. See Extended Data Fig. 6.

L66 - However, the empirical evidence provided is for niche amplitude related to precipitation. It would be important to provide evidence for thermal amplitudes as well.

RESPONSE: We are unaware of any reference showing niche amplitude for temperature in Africa, if the reviewer has a suggestion, please let us know. Because of this, we added the new analysis in ED Fig. 6 which supports our expectations.

L69 - Same comment, what difference? The suggested differences in species susceptibility to climate change are unclear and have not been documented.

RESPONSE: Sentence removed.

L85 - Here, both methodological approaches should be presented: abundance and basal area. In my opinion, reporting basal area results makes it more comparable to previous findings from SA and provides a link with carbon dynamics, as reported by Duque et al. (2021). Shouldn't it be more accurate to use basal area instead?

RESPONSE: We now report both approaches in the main text and discuss the implications of both approaches.

L96 - According to Supplementary Table 1 in the Δ CTI the total number of plots with positive values were 11 (64.7%) and in the Δ CTI basal area 9 (52.9%). Please clarify and refer to supplementary table 1 in this sentence, not just figure 3a.

RESPONSE: We now modified the results to discuss both abundance and basal-area weighted CTI.

L98 - Is this a bootstrapped mean? The values reported in supplementary table 1 differ (0.0038 and 0.0013 for abundance and basal area approaches, respectively).

RESPONSE: The difference came because we had used Worldclim climate in Supplementary Table 1, sorry. This has now been corrected. This is the raw mean, with bootstrapping now used to obtain 95% CIs.

L100 - Supplementary Table 1 names this site as Bwindi. Please use the same name consistently throughout the manuscript.

RESPONSE: corrected.

L101 - values differ from those listed in suppl. table 1. Please clarify.

RESPONSE: Now corrected. See our earlier responses.

L106 - the CPI individual values of each plot should be included in Supplementary Table 1.

RESPONSE: Added as suggested.

L342 - specify dbh threshold

RESPONSE: threshold added, it is ≥ 10 cm diameter

L378 - This is a rather coarse grain pixel for tropical mountains. Suggest to downscale to 90 m² using the SRTM as done before in the Andes: <https://doi.org/10.1111/jbi.13759> or to 30-m using a geographically weighted regression (GWR) model: <https://doi.org/10.1038/s41586-018-0715-9>

RESPONSE: Please see our earlier response (to R3 major comment 1).

L392 - Does the CHELSA precipitation data work well for a 1-ha plot size in the African highlands? It seems that in the Andes no so much: Manz et al. (2016): <https://doi.org/10.1002/2015JD023788>; Buytaert et al. (2010): <https://doi.org/10.5194/hess-14-1247-2010>; Chimborazo & Vuille (2021): <https://doi.org/10.1007/s00704-020-03483-y>

RESPONSE: We appreciate these concerns, and agree that precipitation is less well characterised at fine scales than temperature. For this reason have moved the CPI analysis to supplementary information.

L413 - With high uncertainty in the spatial precipitation records over Africa, these broad-scale precipitation trends are still valid to link with the effect on local tree plot dynamics? See for instance: <https://doi.org/10.1002/2017RG000574>
<https://doi.org/10.1002/joc.4346>

RESPONSE: See our response above. We have added a caveat saying the absence of a convincing relationship between plot precipitation and CPI may reflect the uncertainty in fine-scale precipitation patterns (see lines 142-145).

L422 - verify acronym name

RESPONSE: The 13 was a typo. Now corrected to Delta CTI jackknife.

L450 - In a strict sense, biomass does not die. Suggest using tree mortality or other similar term.

RESPONSE: corrected

L565 - add fit values: r², RMSE, p

each regions shows opposite trends or no trend (UDZ)

RESPONSE: We now report the fit values etc for the relationship between CPI and mean annual precipitation in the main text: "In contrast to CTI, a community precipitation index (CPI, calculated in the same way as CTI, but using the mean annual precipitation in locations species were recorded at) was not significantly related to the mean annual precipitation at each plot (slope = 0.251 ± 0.208 SE, $t = 1.21$, $P = 0.245$, $R^2 = 0.089$, Extended Data Fig. 2)."

Note – there were a few lines highlighted with no comment, so we are unsure what the reviewer was wishing to comment on.

REVIEWERS' COMMENTS

Reviewer #2 (Remarks to the Author):

The authors have successfully addressed my most major concerns on the previous version of their manuscript. I am happy with most of the changes that were made.

Below, I provided a list of additions point that should merit extra attention. I consider these points as minor.

I wish the author best of luck finalizing their work, from which I believe it will be an added value to climate change research in tropical areas.

-general comment: would it be better to convert the unit of thermophilization rates to ‘°C per Decade’, given the very low numbers?

-L16-18: suggested shorter rephrasing: mean rates of thermophilization varied between 0.0023 °C y-1 and 0.0086°C y-1.

L19: suggested rephrasing: ‘were important drivers of thermophilization in Afromontane tree communities.’

L23: I suggest to remove the words ‘predicted and observed’, as there is now a general consensus that species shift poleward and upslope.

L25 and several other locations in the MS: suggest to replace ‘migration’ to range shifts’, as migration refers to the demographic process, and only partly capture the processes behind rang shift (colonization, extinction, changes in abundance etc.)

L33: ad ‘the’ before community temperature index

L33-35: I suggest to not mention the specific rates and regions, just adding the references there is OK I believe.

L42: add ‘local’ to ‘high risk for local species losses?’

L59: suggest to replace ‘resistant’ to ‘resilient’.

L61: ‘migration’ -> range shifts, species’ responses to warming temperatures, ...

L65-66: meaning of the second part of the sentence is not clear. ‘..., might further decrease thermophilization rates in Afromontane tree communities?’

L69: I suggest to remover ‘cooler and warmer- affiliated species here, is this statement applies on any mortality-regeneration dynamics.

L71: Could you add a references to the statement that species from lower elevations have inherent faster growth rates?

L73: remove 'while'?

L93: the subscripts A and B related to CTI are not very informative. Maybe use 'density' and BA as subscripts? Also, here you use capitals, but in the figure lower cases are used.

L100: Could you quantify this association using a correlation coefficient?

Fig2: perhaps good to add correlation coefficients to the plot.

Fig2: 'Abundance-weighted' is not very informative. Basal area is also a measure of abundance. Density? Occurrence?

L107: Do you mean Fig4 here? Are Fig3 and 4 swapped?

L119: rates of temperature changes: this is a bit vague. Is this based on annual means, maxima, minima? Please specify.

L126: referring to Fig4?

L122-126: in my opinion this paragraph can be deleted.

L139-145: I suggest to provide this paragraph as supplementary text, but not in the main text. You've not introduced temporal patterns of precipitation, nor related hypotheses to alterations in precipitation, so it is a bit confusing to mentions these results here. I suggest to focus the main text only on thermophilization trends. Your results are strong enough, I believe.

L162: See my previous comment on the name 'abundance-weighted'. I suggest to look for a more specific term (however I'm not sure what this should be).

L217: delete 'from happening'

L221: referring to CTI-A or CTI-B?

L233-235: can be, but this remains very speculative. I would certainly add caution to this statement and not present it as the most parsimonious explanation. Considerer removing the sentence?

L236: add 'on carbon stocks' after thermophilization?

L242: 'sink strength': did you make a comparison with historical data? If not, this conclusion can't be made from your results.

L244: to make in more general, I suggest to replace 'in a warmer world' to 'under several environmental changes', or something in that line.

L615: is it possible to provide the DOI link to all GBIF records that you extracted?

Extended data fig 6: the density graphs show also possible negative values (especially for Africa), but how is this possible? These should be all >0 ?

Reviewer #3 (Remarks to the Author):

The authors have done a fine job incorporating the majority of the observations and suggestions from the reviewers, including my own. I have no further comments at this point.

REVIEWERS' COMMENTS

Reviewer #2 (Remarks to the Author):

The authors have successfully addressed my most major concerns on the previous version of their manuscript. I am happy with most of the changes that were made.

Below, I provided a list of additions point that should merit extra attention. I consider these points as minor.

I wish the author best of luck finalizing their work, from which I believe it will be an added value to climate change research in tropical areas.

-general comment: would it be better to convert the unit of thermophilization rates to '°C per Decade', given the very low numbers?

We thank the reviewer for this suggestion, but we prefer to keep the units as they are (per year) so they are comparable with previous studies on the topic. Those studies also had low numbers.

-L16-18: suggested shorter rephrasing: mean rates of thermophilization varied between 0.0023 °C y⁻¹ and 0.0086°C y⁻¹.

We thank the reviewer for this suggestion, but we think it is better to keep the units using a + sign so our presentation is consistent with previous studies on the topic.

L19: suggested rephrasing: 'were important drivers of thermophilization in Afromontane tree communities.'

Corrected.

L23: I suggest to remove the words 'predicted and observed', as there is now a general consensus that species shift poleward and upslope.

Another reviewer asked us to keep both terms (predicted and observed), so we prefer to keep them both.

L25 and several other locations in the MS: suggest to replace 'migration' to range shifts', as migration refers to the demographic process, and only partly capture the processes behind range shift (colonization, extinction, changes in abundance etc.)

Corrected

L33: ad 'the' before community temperature index

Corrected

L33-35: I suggest to not mention the specific rates and regions, just adding the references there is OK I believe.

We prefer to show specific rates as later in the manuscript we compare out rates with those mentioned here.

L42: add 'local' to 'high risk for local species losses'?

Corrected

L59: suggest to replace 'resistant' to 'resilient'.

Corrected

L61: 'migration' -> range shifts, species' responses to warming temperatures, ...

Corrected

L65-66: meaning of the second part of the sentence is not clear. '...', might further decrease thermophilization rates in Afromontane tree communities?

Corrected

L69: I suggest to remove 'cooler and warmer- affiliated species here, is this statement applies on any mortality-regeneration dynamics.

We think these terms are important as we are talking about the relative mortality and recruitment of species with different thermal affiliations.

L71. Could you add a references to the statement that species from lower elevations have inherent faster growth rates?

Numerous species from 'lowland' tropical forests have faster growth rates than most species whose range are limited to tropical montane forests (at least in Africa), but not all. This statement is clearly phrased as an expectation (if recruits...) rather than an observation. Determining whether this expectation is met would require extensive examples and references, turning this into a very long paragraph or series of paragraphs. As the statement is just a hypothesis, we prefer not to extend the text and add references here.

L73: remove 'while'?

While is important to the argument of the sentence – we are saying that the stocks are known but the dynamics are not.

L93: the subscripts A and B related to CTI are not very informative. Maybe use 'density' and BA as subscripts? Also, here you use capitals, but in the figure lower cases are used.

We changed A to Stem so that it refers to stem density, and changed B to BA as the reviewer suggested.

L100: Could you quantify this association using a correlation coefficient? and Fig2: perhaps good to add correlation coefficients to the plot.

RESPONSE: In some ways we feel doing this would be odd, as we already tested the bivariate relationships with a linear model and present the slopes, test statistics, p values and R2 of that model. However, we also see that there is some value (e.g. for a future meta-

analysis) in presenting an alternative measure of effect size, so have added the correlation coefficients to the figure legend.

Fig2: 'Abundance-weighted' is not very informative. Basal area is also a measure of abundance. Density? Occurrence?

We clarified we refer to stem abundance here and throughout the manuscript. We use the notation CTI_{Stem} in the figure legends.

L107: Do you mean Fig4 here? Are Fig3 and 4 swapped?

Yes, they were, this has been corrected, thank you.

L119: rates of temperature changes: this is a bit vague. Is this based on annual means, maxima, minima? Please specify.

Corrected – it is based on annual means.

L126: referring to Fig4?

RESPONSE: Figures reordered (see L107 comment) so this reference is now correct.

L122-126: in my opinion this paragraph can be deleted.

We think it is better to clarify here, but if editor thinks it is better to delete, we can do it.

L139-145: I suggest to provide this paragraph as supplementary text, but not in the main text. You've not introduced temporal patterns of precipitation, nor related hypotheses to alterations in precipitation, so it is a bit confusing to mention these results here. I suggest to focus the main text only on thermophilization trends. Your results are strong enough, I believe.

We thank the reviewer for this suggestion, but the recent work of one coauthor in the Americas showed how it is also important to consider potential changes in precipitation, that is why we prefer to mention this here in main text, to avoid the reader thinking that the changes observed are driven by changes in precipitation.

L162: See my previous comment on the name 'abundance-weighted'. I suggest to look for a more specific term (however I'm not sure what this should be).

We clarified by changing term to stem abundance.

L217: delete 'from happening'

Corrected

L233-235: can be, but this remains very speculative. I would certainly add caution to this statement and not present it as the most parsimonious explanation. Considerer removing the sentence?

We changed parsimonious to possible, we believe it is important to provide a potential explanation here, and supplementary data supports such statement.

L236: add 'on carbon stocks' after thermophilization?

Corrected

L242: 'sink strength': did you make a comparison with historical data? If not, this conclusion can't be made from your results.

Mention of sink strength removed

L244: to make in more general, I suggest to replace 'in a warmer world' to 'under several environmental changes', or something in that line.

As the paper is on thermophilization, we prefer to end the main text discussing warmer world, as it is narrower and more related to what we studied that 'environmental change'.

L615: is it possible to provide the DOI link to all GBIF records that you extracted?

We downloaded each species separately, and for those with synonyms, several times, that is why we cannot provide one DOI here.

Extended data fig 6: the density graphs show also possible negative values (especially for Africa), but how is this possible? These should be all >0?

The possible negative values are due to kernel density estimation not treating niche breadth as a bounded space. We could explore using bounded density estimation if the editor thinks this is important, but we would need more time for the review.

Reviewer #3 (Remarks to the Author):

The authors have done a fine job incorporating the majority of the observations and suggestions from the reviewers, including my own. I have no further comments at this point.

Thank you.